# The Association between Parkinson’s Disease and Congestive Heart Failure in Korea: A Nationwide Longitudinal Cohort Study

**DOI:** 10.3390/jpm13091357

**Published:** 2023-09-05

**Authors:** Jimin Kim, Hakyung Kim, Sol Bi Kim, Woo Yup Kim, Seung Hun Sheen, Inbo Han, Je Beom Hong, Seil Sohn

**Affiliations:** 1Cornell University, Ithaca, NY 14850, USA; jk2497@cornell.edu; 2Genome & Health Big Data Branch, Department of Public Health, Graduate School of Public Health, Seoul National University, Seoul 08826, Republic of Korea; alicekm@snu.ac.kr; 3Department of Neurosurgery, CHA University College of Medicine, 59, Yatap-ro, Bundang-gu, Seongnam-si 13496, Republic of Korea; kimsolb5@naver.com (S.B.K.); wooyeabkim@gmail.com (W.Y.K.); nssheen@gmail.com (S.H.S.); hanib@cha.ac.kr (I.H.); 4Department of Neurosurgery, Kangbuk Samsung Hospital, Sungkyunkwan University School of Medicine, Seoul 03181, Republic of Korea; jebeomhong@gmail.com

**Keywords:** Parkinson’s disease, population, congestive heart failure, epidemiology

## Abstract

The purpose of this nationwide longitudinal follow-up study is to investigate the relationship between Parkinson’s disease (PD) and congestive heart failure (CHF) patients in Korea. Patient data were collected using the National Health Insurance Service (NHIS) Health Screening (HEALS) cohort. The International Classification of Diseases 10-CM code G-20 distinguished 6475 PD patients who were enrolled in the PD group. After removing 1039 patients who were not hospitalized or attended an outpatient clinic less than twice, the total number of participants was reduced to 5436 individuals. Then, 177 patients diagnosed before 1 January 2004 were removed for relevancy, leaving us with 5259 PD patients. After case–control matching was completed using 1:5 age- and gender-coordinated matching, 26,295 people were chosen as part of the control group. The Cox proportional hazards regression analysis and the Kaplan–Meier technique were used to assess the risk of CHF in patients with Parkinson’s disease. After controlling for age and gender, the hazard ratio of CHF in the PD group was 5.607 (95% confidence interval (CI), 4.496–6.993). After that, the hazard ratio of CHF in the PD group was modified against for comorbid medical disorders, resulting in a value of 5.696 (95% CI, 4.566–7.107). In subgroup analysis, CHF incidence rates were significantly increased in the PD group compared to the control group (males and females; aged ≥ 65 and <65; the non-diabetes and diabetes, hypertension and non-hypertension, and dyslipidemia and non-dyslipidemia subgroups). This nationwide longitudinal study shows a higher incidence rate of CHF in PD patients.

## 1. Introduction

Parkinson’s disease (PD) is the second most common neurodegenerative disease that affects human movement over time, affecting 1% to 3% of people aged > 65 years [1]. In the past 30 years, the odds of developing PD have increased to 2.5 times as much as before, and more than 6 million people are diagnosed annually worldwide [2]. The main characteristic of this disease is the early prominent death of dopaminergic neurons in the substantia nigra pars compacta [3]. The pathological characteristic is neural inclusions in the form of Lewy bodies and neurites, as well as cell loss in the substantia nigra and other areas of the brain. Motor symptoms of PD include bradykinesia, rigidity, and rest tumor, as well as changes in posture and gait [4]. In addition, there are non-motor symptoms such as hyposmia, constipation, urinary dysfunction, orthostatic hypotension, memory loss, depression, pain, and sleep disturbances.

Currently, congestive heart failure (CHF) is the leading cause of death. Due to the increase in life expectancy and developments in treatment methods, the number of diagnoses is gradually increasing. Notable symptoms include dyspnea on exertion, peripheral edema, orthopnea, and paroxysmal nocturnal dyspnea [5].

Past studies have found that PD can have critical effects on the heart, as well as the blood vessels [6]; recent research indicates that PD is associated with hypertension, diabetes, and other vascular risk factors [7,8,9]. A longitudinal cross-sectional study revealed that elderly PD patients were twice as likely to have heart failure than non-PD patients [10]. We want to investigate PD’s relationship with a specific type of cardiovascular disease (congestive heart failure) in the Korean population through the Cox proportional hazards regression analysis and Kaplan–Meier technique. This countrywide longitudinal research has been adjusted to account for the following variables: age, gender, diabetes, hypertension, and dyslipidemia. The aim of this study is to identify the risk of CHF in PD patients located in Korea.

## 2. Materials and Methods

### 2.1. Data Source

Eight researchers from three different institutions in South Korea gathered data on PD patients from the National Health Insurance Sharing Service (NHIS), a single-payer health system in South Korea, from 2004 to 2015. This particular cohort is credible and reliable as healthcare providers are required to submit medical claims to the NHISS for further review; furthermore, the system itself requires individuals aged ≥ 40 years to take national health examinations, twice every year for office employees and once every year for non-office employees. To elaborate, the data incorporate details about the demographics of individuals, as well as different medical treatments, procedures, and diagnoses they received, as followed by the 10th revised codes of the International Classification of Diseases (ICD-10). We acquired the rights to use the NHIS database from the institutional review board (IRB No. 2020-01-011). Prior consent was not necessary because the data used in this study were anonymous secondary data that were made available for research purposes.

### 2.2. Study Design and Subjects

Following sex- and age-stratified matching intended to determine the potential dangers of CHF in PD patients, the population incorporating both the PD and control groups comprised 515,547 patients (Figure 1). The Korea NHISS database collected information from the participants for up to 12 years, from 1 January 2004 to 31 December 2015. In this cohort, the patients’ demographics and medical history were utilized, which include sex, age, disease codes (using the ICD-10 codes), and income level; additionally, the risk of CHF was estimated after the database was modified for sex, age, and comorbidities such as hypertension, dyslipidemia, and diabetes mellitus. Information on such comorbidities were analyzed through the NHISS’ inpatient and outpatient records.

### 2.3. Establishment of the Study Cohort

Out of the 515,547 patients in the NHISS cohort, the number of PD patients was 6475, distinguished by code G20 on the ICD-10 code. Out of these, 5436 patients who went to the outpatient clinic more than twice or who were hospitalized more than once were chosen. Again, the population decreased after those with preexisting PD were removed, leading to a final total of 5259 patients who were recently diagnosed (after 1 January 2004). On the other hand, the population of the control group was determined using the R package match algorithm ‘Match IT’ and 1:5 age- and sex-stratified matching without replacement, which resulted in 26,295 people [11,12]. Data were gathered from both the control and PD groups until 31 December 2015.

### 2.4. Statistical Analysis

The mean differences in the PD and control groups’ demographics and comorbidities were compared through Student’s *t*-test and the chi-square test. The CHF-free survival rates of both categories were determined through the Kaplan–Meier method, while the differences in disease-free survival were estimated utilizing Wilcoxon’s log rank test. Additionally, approximations on the impact of PD on respective events were tested using multivariate analyses with two Cox proportional hazards regression models. There were three types of Cox proportional hazards regression models that were used: while sex and age were modified in model 1, age, sex, and income were regulated in model 2, and age, sex, income, and other diseases were regulated in model 3. The Cox proportional hazards regression model was used to determine the association between PD and each respective event in each subgroup. The following analyses were deduced using R software (version 3.3.3).

## 3. Results

### 3.1. Characteristics of the PD and Control Group

There were 5259 subjects in the PD group and 26,295 in the control group. Table 1 shows the different characteristics of the participants: sex, age, low income, diabetes, hypertension, and dyslipidemia. The prevalence of diabetes was higher in the PD group than the control group, with statistical significance (15.73% vs. 13.99%, *p* = 0.001). However, data for other variables (hypertension, low income) showed higher prevalence in the control group, with statistical significance (46.96% vs. 44.46%, *p* < 0.001 for hypertension, 24.81% vs. 27.23%, *p* < 0.001 for low income). Dyslipidemia was higher in the control group (17.80%) than the PD group (16.77%). However, it was statistically insignificant (*p* = 0.077).

### 3.2. CHF in PD and Control Group

The incidence rate of CHF was higher in those with PD (7.496) than the control group (1.575). The data were then analyzed through three models. Model 1 was adjusted for age and sex, model 2 was adjusted for age, sex, and income, and model 3 was adjusted for age, sex, income, diabetes, hypertension, and dyslipidemia. After these analyses, the data showed that the PD group was five times more likely to be at risk for CHF. The Cox proportional hazards regression model indicated that the CHF hazard ratio of the PD group for model 1 is 5.607 (95% CI, 4.496–6.993, Table 2), for model 2 is 5.696 (95% CI, 4.566–7.107, Table 2), and for model 3 is 5.668 (95% CI, 4.544–7.071, Table 2). The risk of CHF in the PD group was dramatically higher than in the control group, according to the Kaplan–Meier curves (Figure 2).

### 3.3. Subgroup Analysis of CHF Incidence Rate

To further examine CHF’s incidence rate in the PD group, a subgroup analysis was performed according to age, sex, diabetes, dyslipidemia, and hypertension. In all subgroups, the incidence rates of CHF in the PD group were shown to be higher than the control group (Table 3).

## 4. Discussion

Around 1~3% of individuals aged ≥ 65 years are diagnosed with PD [13]. As the average life span increases, more people are found to have this disease [14]. PD is associated with increased mortality [15] and CHF is one of the leading cause of death. However, the exact relationship between the two illnesses is yet to be researched.

In this extensive and comprehensive large-scale cohort study conducted on a nationwide scale, meticulously designed to include age- and sex-matched participants, we have successfully identified a significant association between PD and the occurrence of CHF. The results of our analysis indicate that patients who have been diagnosed with PD exhibit an elevated and considerable risk of developing CHF.

One population-based follow-up study from Taiwan reported that the rate of CHF in a PD group was 2.4 times higher than in a control group [16]. Similarly, Park et al. reported a high association between cardiovascular diseases and PD, with a hazard risk of 1.65 (95% CI, 1.52–1.78) [13] In our research, the incidence rate of CHF in the PD group was 7.496, which is in agreement with previous studies. Additionally, through subgroup analysis, this study was able to confirm that PD causes an increased risk of CHF without consideration of age, sex, DM, hypertension, and dyslipidemia. One of the advantages of this large-scale cohort study was having the opportunity to cover more than 97% of the entire Korean population through the National Health Insurance Service. This cohort tracks all medical records, which enabled us to correctly evaluate the temporal relationship between PD and CHF. A reliable uniform diagnostic criterion for PD was also used.

There have been many attempts to prove the relationship between PD and other diseases, such as AMI and ischemic stroke [17,18]. However, definite pathophysiological evidence has yet to be found. The association between ischemic stroke, AMI, and PD is surmised to be due to the oxidative stress in PD patients causing damage to the dopaminergic cells of the substantia nigra [19,20]. As oxidative stress accumulates in the endothelial cell, it likely results in atherosclerotic change, which increases the risk of ischemic stroke. Also, studies indicate that PD patients can also develop orthostatic hypotension, which can cause ischemic brain parenchymal damage [21].

In this study, we tried to find the relationship between PD and CHF. Though the mechanism of an increased risk of CHF in PD patients is not certain, one reasonable theory surmises that the specific areas of the brain that control the cardiovascular system have encountered neurodegeneration in PD [22]. PD patients were found to have decreased parasympathetic activity and increased sympathetic activity. Such imbalance of both tonicities causes a higher risk of cardiac damage though tachycardia and vasoconstriction [22]. PD patients can also have synucleinopathy—a neurodegenerative disease that is known for its abnormal amount of nerve fibers, glial cells, or aggregates of alpha-synuclein protein in neurons [23]. Due to synucleinopathies in PD, cardiac sympathetic denervation may occur, resulting in failing to increase the low blood pressure that might be caused by the baroreceptor function failure to communicate with the heart and blood vessels [23]. Such a behavior is related to baroreflex failure, extracardiac noradrenergic denervation, sympathetic denervation, and orthostatic hypotension [24,25,26].

Another reason behind the association between PD and CHF is heart rate variability: recent studies have found that patients who developed PD had more decreased heart rate variability [27]. Other previous research found that PD patients have a prolonged PR interval and QTc interval on their electrocardiogram [28]. This is often caused by oxidative stress and inflammation. Certain studies showed that systemic inflammation and oxidative stress both interfere with cellular functions and hence are related to atherosclerosis development [19,20,29].

Limitations of this study are also worth mentioning. For example, information about certain inflammatory markers (interleukin-6 and C-reactive protein) and other possible confounding variables (smoking habits, alcohol consumption, physical exercise, body mass index, family history, PD severity, medical records, imaging data) are not present in this study, making it more complex to analyze the novel effects on the association between CHF and PD. However, though there are limitations, it is undeniable that this is the first nationwide longitudinal follow-up study that describes the relationship between CHF and PD in South Korea. The results of this research helped to further fathom the association between PD and CHF, as well as its effects. Similar studies are expected to improve patient care in the medical and health sector.

## 5. Conclusions

This nationwide longitudinal cohort study deduced that PD patients residing in Korea, in general, have an increased risk of CHF. Consequently, this study implies that the risk of CHF should be considered during patient care in those diagnosed with PD.

## Figures and Tables

**Figure 1 jpm-13-01357-f001:**
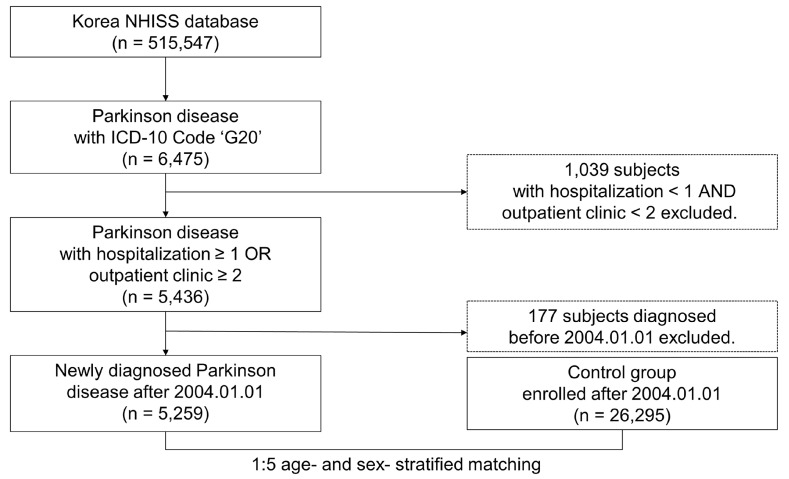
This flow chart, which supports this 12-year cohort study using the NHISS cohort, shows the data creation procedure. NHISS: National Health Insurance Sharing Service, ICD-10: International Classification of Diseases 10.

**Figure 2 jpm-13-01357-f002:**
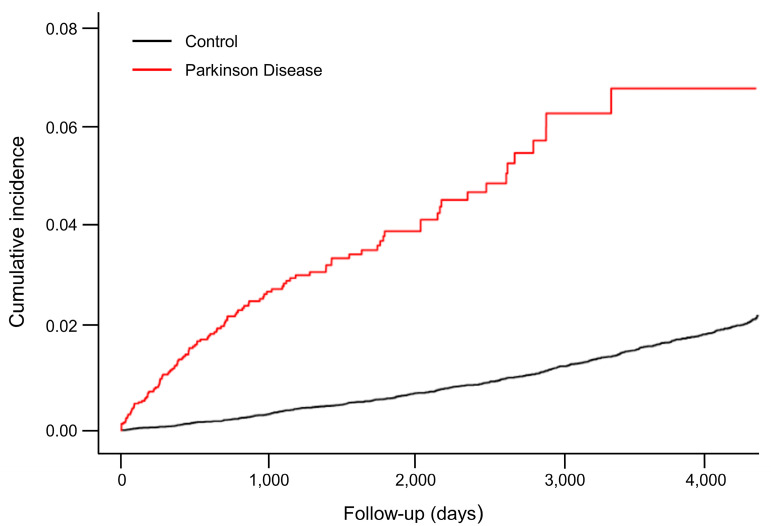
Parkinson’s disease (PD) and control groups’ cumulative rates of congestive heart failure (CHF) were compared. Kaplan–Meier curves were used to analyze the cumulative risks of CHF in the PD and control groups.

**Table 1 jpm-13-01357-t001:** Characteristics of PD and control group.

Variables	Control (*n* = 26,295)	PD (*n* = 5259)	*p*-Value
Male (%)	14,010 (53.28)	2802 (53.28)	1
Age (mean (SD))	62.87 ± 8.42	62.87 ± 8.42	1
Age ≥ 65 (%)	12,515 (47.59)	2503 (47.59)	1
Low income (%)	7159 (27.23)	1305 (24.81)	<0.001 *
Diabetes (%)	3678 (13.99)	827 (15.73)	0.001 *
Hypertension (%)	12,349 (46.96)	2338 (44.46)	<0.001 *
Dyslipidemia (%)	4681 (17.80)	882 (16.77)	0.077

* Indicates statistical significance. PD, Parkinson’s disease.

**Table 2 jpm-13-01357-t002:** Adjusted hazard ratio for CHF event.

Disease	PD	N	Event	Duration (Days)	Duration (Years)	Incidence Rate (%)	Hazard Ratio (95% CI)		
							Model 1	Model 2	Model 3
CHF	0	26,295	435	100,821,984	276,224.614	1.575	1 (Ref.)	1 (Ref.)	1 (Ref.)
	1	5259	120	5,843,469	16,009.504	7.496	5.607 [4.496, 6.993]	5.696 [4.566, 7.107]	5.668 [4.544, 7.071]

Model 1: adjusted for age and sex; model 2: adjusted for age, sex, and income; model 3: adjusted for age, sex, income, diabetes, hypertension, and dyslipidemia. PD, Parkinson’s disease; CHF, congestive heart failure; CI, confidence interval.

**Table 3 jpm-13-01357-t003:** Subgroup analyses between PD and control group.

Variables	Control		PD		Hazard Ratio (95% CI)
	Event	Incidence Rate (%)	Event	Incidence Rate (%)	
Sex					
Male	156	1.256	48	7.006	6.908 [4.854, 9.832]
Female	279	1.835	72	7.861	5.017 [3.780, 6.660]
Age					
<65	81	0.531	31	3.628	6.762 [4.286, 10.670]
≥65	354	2.863	89	11.924	5.386 [4.180, 6.938]
Diabetes					
No	358	1.501	99	7.325	4.378 [2.603, 7.362]
Yes	77	2.044	21	8.418	5.996 [4.698, 7.654]
Hypertension					
No	212	1.433	54	6.130	4.934 [3.563, 6.832]
Yes	223	1.738	66	9.165	6.527 [4.300, 8.820]
Dyslipidemia					
No	374	1.653	105	7.890	5.764 [4.547, 7.307]
Yes	61	1.221	15	5.553	5.088 [2.774, 9.332]

PD, Parkinson’s disease; CI, confidence interval.

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
