# Peer review of "The Association between Parkinson’s Disease and Congestive Heart Failure in Korea: A Nationwide Longitudinal Cohort Study"

_jpm, 2023, doi:10.3390/jpm13091357_

Round 1

Reviewer 1 Report

The authors tried to find the relationship between PD and CHF; the discussion seems to me to be too vague on such a difficult topic It is a very broad topic, the work is based on a great deal of material, which is its value. Theis longitudinal study showed a higher incidence rate of CHF in PD patients.

Author Response

We sincerely appreciate your time and effort in reviewing our paper. Your review comments have been incredibly valuable in helping us improve the manuscript. We have made revisions based on the comments you provided. We kindly request another review of the revised content. I sincerely appreciate your assistance. Thank you so much.  

The authors tried to find the relationship between PD and CHF; the discussion seems to me to be too vague on such a difficult topic 

  •  Thank you so much. We have made additional modifications to the discussion section and indicated the changed parts in the main text using different colors. 

Sincerely,

Jessica Kim 

Reviewer 2 Report

Manuscript shows the prevalence of Congestive Heart Failure in Parkinson patients troughs 12 years in Korea. I have some considerations for you.

Introduction

Introduction is too small, more explanation about the topic should be added.

Methodology

Who collected the data? How many researchers took part in the study? Where? Some information about the procedure must be explained.

Results

In table 3, N and Y should be explained on the footer.

The article is too brief, there much more information about the topic that authors could be addressed. I am not sure if the manuscript provides any new information, please complete them to decide.

Author Response

We sincerely appreciate your time and effort in reviewing our paper. Your review comments have been incredibly valuable in helping us improve the manuscript. We have made revisions based on the comments you provided. We kindly request another review of the revised content. I sincerely appreciate your assistance. Thank you so much.  

Introduction - Introduction is too small, more explanation about the topic should be added. 

  • Thank you so much. We have made additional modifications to the introduction section and indicated the changed parts in the main text using different colors. 

Methodology - Who collected the data? How many researchers took part in the study? Where? Some information about the procedure must be explained. 

  • We added the following sentence.  
  • Eight researchers from three different institutions in South Korea.” 
  • And other informations were already described in the methods section. 

Results 

In table 3, N and Y should be explained on the footer. 

  • We have changed 'N' to 'no' and 'Y' to 'yes'. 

The article is too brief, there much more information about the topic that authors could be addressed. I am not sure if the manuscript provides any new information, please complete them to decide. 

  • Thank you for your excellent feedback. We have added additional content to the overall text. The modified sections are indicated in a different color. We sincerely appreciate your assistance. 

Sincerely,

Jessica Kim 

Round 2

Reviewer 2 Report

Dear authors, 

Manuscript looks much better. Discussion has improved its quality.

Thank you